# In Vitro Evaluation of Acute Toxicity of Five *Citrus* spp. Essential Oils towards the Parasitic Mite *Varroa destructor*

**DOI:** 10.3390/pathogens10091182

**Published:** 2021-09-13

**Authors:** Roberto Bava, Fabio Castagna, Cristian Piras, Ernesto Palma, Giuseppe Cringoli, Vincenzo Musolino, Carmine Lupia, Maria Rosaria Perri, Giancarlo Statti, Domenico Britti, Vincenzo Musella

**Affiliations:** 1Department of Health Sciences, University of Catanzaro Magna Græcia, CIS VetSUA, Viale Europa, 88100 Catanzaro, Italy; roberto.bava@unicz.it (R.B.); fabiocastagna@unicz.it (F.C.); palma@unicz.it (E.P.); britti@unicz.it (D.B.); musella@unicz.it (V.M.); 2Nutramed S.c.a.r.l. Complesso Ninì Barbieri, Roccelletta di Borgia, 88021 Catanzaro, Italy; 3Institute of Research for Food Safety & Health (IRC-FISH), Department of Health Sciences, University of Catanzaro Magna Græcia University, 88100 Catanzaro, Italy; 4Department of Veterinary Medicine and Animal Production, University of Naples Federico II, CREMOPAR Campania Region, 80137 Naples, Italy; cringoli@unina.it; 5Mediterranean Etnobotanical Conservatory, Sersale, 88054 Catanzaro, Italy; studiolupiacarmine@libero.it; 6Department of Pharmacy, Health and Nutritional Sciences, University of Calabria, 87100 Cosenza, Italy; mariarosaria.perri@unical.it (M.R.P.); g.statti@unical.it (G.S.)

**Keywords:** *Apis mellifera*, *Varroa destructor*, Calabria region, *Citrus* essential oils, anti-parasitic efficacy, green veterinary pharmacology, in vitro toxicity

## Abstract

*Varroa destructor* is the most important ectoparasitic mite of honey bees that has a negative impact on bee health and honey production. The control programs are mainly based on the use of synthetic acaricides that are often administered indiscriminately. All this has led to drug resistance that now represent a great concern for honey bee farming. The research for alternative products/methods for mites’ control is now mandatory. The aim of this study was to test whether *Citrus* spp. essential oils could diminish the growth of the *V. destructor* mite. In Calabria (southern Italy), plants of the *Citrus* genus are very common and grow both spontaneously and cultured. The essential oils used in this study were extracted from bergamot (*Citrus bergamia*), grapefruit (*Citrus paradisi*), lemon (*Citrus limon*), orange (*Citrus sinensis*), and mandarin (*Citrus reticulata*) by hydrodistillation. Every EO was in vitro tested against *V. destructor*. Each experimental replicate was performed using 35 viable adult female mites (5 for each EO) collected the same day from the same apiary and included negative controls (5 individuals exposed to acetone only) and positive controls (5 individuals exposed to Amitraz diluted in acetone). The essential oils (Eos) were diluted (0.5 mg/mL, 1 mg/mL, and 2 mg/mL) in HPLC grade acetone to obtain the working solution to be tested (50 µL/tube). Mite mortality was manually assessed after 1 h exposure under controlled conditions. The essential oils that showed the best effectiveness at 0.5 mg/mL were bergamot, which neutralized (dead + inactivated) 80% (*p* ≤ 0.001) of the parasites; grapefruit, which neutralized 70% (*p* ≤ 0.001); and lemon, which neutralized 69% of them. Interestingly, the positive control (Amitraz) at the same concentration neutralized 60% of the parasites. These results demonstrate that Calabrian bergamot, grapefruit, and lemon Eos consistently reduced *V. destructor* viability and open the possibility for their utilization to control this parasite in honey bee farming.

## 1. Introduction

Bees are essential for the regulation of ecosystems, and they are among the most economically important pollinators for crops and wild plants [1,2,3]. It is widely known that most of the world’s wild flower plantpopulation (87.5%) and one-third of the crops used for human consumption depend on animal pollination (especially bees) for sexual reproduction [4,5,6]. Equally, it is well known that honey bee farming generates income and job opportunities in rural areas and allows human beings to obtain valuable products with relevant nutritional and nutraceutical properties [6,7,8]. 

Recently, this species survivor has been hampered by a constantly increasing loss of honey bee colonies that now is reaching concerning levels [9]. The causes are not yet completely known, but they are certainly linked to different human-related variables/factors, such as intensive agriculture and the use of pesticides [10,11]. This led to consequences such as starvation and malnutrition of honey bees due to the diminished biodiversity and to the invasion of species such as the Asian hornet (*Vespa velutina*), the small hive beetle (*Aethina tumida*), and the parasitic mite *Varroa destructor* [11,12,13]. Currently, the *V. destructor* parasite represents the greatest threat to honey bee health. This parasite has endangered the European honey bee (*Apis mellifera*) with its spread from its native host, the Asian honey bee (*Apis cerana*) [14]. Its presence in farms is negatively affecting profitability, favoring the incidence of pathogens (bacterial and viral), and reducing the number of non-reared hives [15,16,17].

Keeping the parasite levels below the damage threshold is extremely important to maintaining bee farms healthy.

Beekeepers have a wide range of different synthetic acaricides available to control mite populations. The usual choice is often that of synthetic drugs, such as Amitraz, because of their effectiveness and ease of use. However, these pharmacological preparations can contaminate beehive products [18] and have a negative impact on honey bees’ health [19]. In addition, *V. destructor* is rapidly developing drug resistance, which is now increasingly reported. Resistance to the main synthetic acaricides, such as fluvalinate, has been well documented, and drug resistance to Amitraz is becoming widespread [20,21,22,23].

An increasing self-awareness concerning the importance of the environment and a greater concern of consumers for safety of agricultural products are encouraging the development and introduction of management practices and remedies with low environmental impact [24,25]. This is supported by international legislation that is moving in an eco-friendly direction [24,25]. From this perspective, eco-compatible treatments based on natural substances, also present in food, such as organic acids (formic acid, oxalic acid) are already available. However, these substances usually require high dosages/concentrations to be effective and, as a consequence, can cause adverse effects on honey bees [26,27].

Essential oils (EOs) of various officinal plants are good candidates as alternative methods because they have often been already investigated for their eventual toxicity for humans and the environment. Control strategies based on the use of EOs are part of the broad concept of green pesticides or bio-pesticides. The term bio-pesticide refers to all kinds of natural pest control techniques that reduce the pest population while paying attention to food and environmental safety. Substances such as plant extracts, pheromones, hormones, and secondary metabolites of microorganisms are also included in the concept of green pesticides. These approaches, on top of being completely natural, have the advantage of promoting less resistance phenomena [28]. Due to their high number of constituents and the multiple target receptors involved in the mechanism of action, EOs appear to have no specific cellular targets, and treated mite populations are less likely to develop resistance. Therefore, the use of EOs in beekeeping and for the control of honey bee diseases could open the path towards a breeding system based exclusively on green veterinary pharmacology (GVP), with significant benefits for honey bee health and the environment.

Even though *Citrus* (Rutaceae) EOs have been widely investigated for human nutraceutical use, there is still a gap of knowledge about their possible efficacy against honey bee mites. The genus *Citrus* includes numerous species widely cultivated for fruit—namely, *Citrus* fruit. Among those, we can mention the bitter orange (*C. aurantium*), the sweet orange (*C. sinensis*), the grapefruit (*C. maxima*), the pink grapefruit (*C. paradisi*), the lemon (*C. limon*), the mandarin (*C. reticulata*), and the bergamot (*C. bergamia*) [29]. Among their bioactive functions, well-documented effects are reported against a range of bacterial [30,31,32], fungal [33,34], and viral pathogens [35,36]. Furthermore, in the last decades, the toxic properties of different *Citrus* EOs against parasites and harmful insects were explored. In particular, Safavi and Mobki (2016) [37] reported the fumigant toxicity of *C. reticulata* peel essential oil against a stored-product insect pest, *Tribolium castaneum* (Tenebrionidae) larvae. Kumar et al. (2012) [38], using contact toxicity and fumigation bioassays, verified the insecticidal activity of the essential oil of *C. sinensis* against the larvae and pupae of *Musca domestica* (Muscidae). Pazinato et al. (2016) [39] verified the acaricidal efficacy of seven essential oils against cattle ticks. They found that *C. aurantium* oil was effective (73% to 95% of inhibition) on tick deposition (partial or total) and that there was linearity between the concentrations used and the effectiveness.

From a geographical point of view, *Citrus* crop cultivation is mostly confined to Southern Italy, particularly in the Sicily and Calabria regions, supplying 87% and 83% of the national production of the oranges and mandarins, respectively. The Provinces of Cosenza, Catanzaro, and Reggio Calabria represent the areas with the highest citrus production in Calabria. In these regions are produced the greatest quantities of Italian clementines, and this sector represents an important share of the GDP of the agricultural income of the region [40]. Moreover, the Calabria region has a monopoly on some products, such as cedar and bergamot. Agricultural by-products also represent an additional source of income for local industries processing these fruits. The transformation of the waste into marketable products, such as essential oils, could represent a green approach that may add some commercial value to this market.

According to this premise, the in vitro acaricidal effect of five *Citrus* spp. EOs (fruits cultivated and extracted in Calabria) was investigated against the *V. destructor* mite. The varroacidal effect was evaluated at different EO concentrations and, consequently was tested for acute toxicity towards the honey bees.

## 2. Results

### 2.1. Chemical Characterization

As demonstrated in Table 1, for each essential oil, at least 11 compounds (bergamot EO) were detected, with a maximum of 18 compounds detected for orange oil. Alpha-pinene, beta-myrcene, limonene, and linalool were detected in all 5 analyzed essential oils, and the most effective bergamot oil showed the highest concentration of linalyl acetate.

Limonene represented the most abundant compound in each analyzed sample, with the highest percentage (47.85%) both in mandarin and orange essential oils, followed by grapefruit (47.62%); the lowest amount of limonene, instead, was detected in bergamot, with a percentage value of 25.17%. Phellandrene and alpha-terpinolene were detected in the lemon sample only.

### 2.2. Screening Natural Compounds for Their Toxicity to V. destructor

Every essential oil was used at the concentration of 0.5 mg/mL, 1 mg/mL, and 2 mg/mL diluted in acetone. Acetone alone (medium) was used as negative control; the anti-mites drug Amitraz was used as positive control to assess the effectiveness of the anti-parasitic treatment on the same population. For each experiment, five mites were incubated inside a previously filled vial with each essential oil and with acetone or Amitraz as negative and positive control, respectively. The experimental design was adopted from Gashout and Guzmán-Novoa, with some minor modifications concerning the material used [41].

Seven technical replicates with this experimental design were carried out for the 1 mg/mL and 2 mg/mL concentrations. Fourteen replicates were performed to test the lowest concentration of 0.5 mg/mL. In total, 980 mites were tested: 490 for the concentrations of 1 mg/mL and 2 mg/mL and another 490 mites for the lowest concentration of 0.5 mg/mL, divided in all the set of trials.

In Table 2 are reported the percentage values and standard deviation values (SD), of the *V. destructor* parasite being neutralized in each trial at concentrations of 0.5, 1, and 2 mg/mL with different *Citrus* spp. Essential oils, acetone (negative control), and Amitraz (positive control).

Figure 1 represents the evaluation of every essential oil used in this study; the viability of the mites was manually evaluated after 1 h of incubation. The *y* axis represents the percentage of parasites neutralized after 1 h when using 0.5 mg/mL, 1 mg/mL, or 2 mg/mL of the essential oil/drug. The difference between the number of neutralized parasites for each essential oil in comparison with the control (acetone) was found to be extremely significant. This was the case even when the acetone effect was compared with the lowest concentration used (0.5 mg/mL; *p* ≤ 0.001) for the less effective Eos, such as mandarin and orange. Using higher essential oils amounts (1 and 2 mg/mL) did not substantially modify the efficacy. Among the tested oils, bergamot and lemon showed the highest efficacy in neutralizing mites, both at the concentration of 0.5 and 1 mg/mL. It is interesting to point out the higher efficacy of Amitraz when used at a concentration of 2 mg/mL in comparison with 0.5 mg/mL and 1 mg/mL. Such a high concentration of this drug left no parasite alive in any of the seven different experimental replicates. Such efficacy was not reached for any EO, even when it was used at the highest concentration of 2 mg/mL.

Figure 1a represents the comparison between the negative control test performed just with acetone, the positive control test, where parasites were exposed to the anti-mite drug Amitraz at a concentration of 0.5, 1, and 2 mg/mL, and the bergamot (2a), grapefruit (2b), lemon (2c), orange (2d), and mandarin (2e) essential oils used at the same concentration (0.5 mg/mL, 1 mg/mL, and 2 mg/mL).

### 2.3. Screening for Toxicity towards Honey Bees

In order to assess the eventual toxicity of the tested essential oils on honey bees, a pool of five honey bees, randomly chosen, were exposed to the same concentrations of the most effective (bergamot and lemon) essential oils/drugs. After a one-hour exposure, the honey bees were transferred to different containers free of essential oils/drug, and the mortality was evaluated at 24 and 48 h (see Materials and Methods section). The honey bees’ behavior was observed for the following 48 h, and no mortality or illness was reported.

## 3. Discussion

Many strategies have been proposed for the control of the *V. destructor* parasite. The scientific community is now trying to move towards the selection of honey bees resistant to this parasite [42]. This path would represent the most effective and the most independent path of any pharmacological treatment and, in turn, from any resistance phenomenon. However, it may take too long for several reasons. The first one is due to the difficult identification of honey bee populations that consistently adopt hygiene behaviors that are able to counter this parasite growth. The second one is that there is no well-established and regulated market for these honey bee populations in Europe.

For this reason, it is more immediate and beneficial to pursue the path of trying to control this parasitosis through treatment with drugs or natural compounds/remedies. Lately, due to the high resistance phenomena to numerous anti-mite drugs, many “soft” acaricides have been successfully employed for this task [15]. Among those, such organic acids as formic acid, oxalic acid, and lactic acid are used; among EOs, thymol is mostly employed. The employment of “soft” acaricides seems to be very promising because they are less prone to induce resistance mechanisms due to their heterogeneous chemical composition with multiple different functions/molecular targets [15]. However, there are some adverse effects that are worth taking into consideration. For example, the treatment with thymol can lead to adverse reactions including brood mortality and removal [43,44] and can cause mortality in queens, although they are less sensitive to thymol than the workers [45]. Furthermore, the thymol has been found to induce alterations in the behavior of honey bees, such as greater activity and reduced response to light stimuli [46].

The ideal acaricide is, possibly, represented by a natural compound able to selectively counteract *V. destructor* growth without interfering with honey bees’ health/behavior. “Soft” acaricides, such as thymol, are valuable natural resources and have opened the boundaries for their employment in practice. However, there are many other EOs that might be used for this purpose. Following this thread, it has to be mentioned that there is still quite a scientific gap in the knowledge and experimentation of *Citrus* oils for this purpose.

As described in the Introduction section, the Calabrian region is very rich in different *Citrus* fruit species that are commercially used for nutritional, cosmetic, and nutraceutical purposes. This represents a great resource and offers the possibility of using both extracts of the whole fruit and extracts of the parts of fruit typically discarded after the fruit is processed.

The in vitro efficacy of bergamot (*C. bergamia*), grapefruit (*C. paradisi*), lemon (*C. limon*), orange (*C. sinensis*), and mandarin (*C. reticulata*) was tested at different concentrations to measure their effectiveness in inactivating the *V. destructor* parasite.

The essential oils of *Citrus* species were characterized to assess their chemical composition and to exclude any possible variability linked to the different extraction methods and to the geographical area of growth [47]. Every analyzed EO showed a high percentage of limonene. In previous studies, limonene and beta-amino alcohol derivatives have proved to be good candidates for controlling mite infestations [48]. However, lower concentrations of limonene were present in two of the most effective essential oils but were balanced by a higher concentration of beta-pinene, another monoterpene that has been shown to have acaricidal action in several experimental tests [49]. Bergamot oil, which was found to be one of the most effective, showed very high concentrations of linalyl acetate. This compound was found to be present in high amounts in *Lavanda angustifolia* essential oil and is known for its acaricidal activity [50]. Linalool is present in all EOs, but in a very high concentration in bergamot EO. This compound is well documented as well for its strong activity against the *Sarcoptes scabiei* parasite. The collected evidence certainly explains the strong varroacidal effect detected and most probably suggests that it is the combination of the different concentrations of these detected compounds that is contributing to the successful mite inactivation.

All tested oils showed a good level of effectiveness in comparison with the respective control (only acetone), and their efficacy was statistically significant even when used at 0.5 mg/mL (Figure 1a). The essential oils that showed greater efficacy at the lowest concentration of 0.5 mg/mL were bergamot, which, neutralized (dead + inactivated) 80% (*p* ≤ 0.001) of the parasites; grapefruit, which neutralized 70% (*p* ≤ 0.001); and the lemon, which neutralized 69% (*p* ≤ 0.001) of them. Around the same efficacy was visible for higher concentrations of 1 and 2 mg/mL. This in vitro result is extremely relevant considering that Amitraz preparation for “in field” use is at least of 18 g/kg (beehive strips for honey bees).

EOs were tested for their toxicity on honey bees that were exposed at the same concentrations used for the tests on mites. After a one-hour exposure, their behavior was monitored for the following 48 h to measure any visible consequence related to exposure. No mortality or any visible behavioral anomaly was observed. This might suggest the absence of toxicity for honey bees exposed to the same conditions that inactivate mites and lays the groundwork for future investigation of these proposed compounds in the field.

## 4. Materials and Methods

### 4.1. Essential Oils Used for the Experiments

Essential oils of *Citrus bergamia*, *C. sinensis*, *C. limon*, *C. reticulata*, and *C. paradisi* were used in the experiments. Essential oils furocoumarins-free (FCF) were purchased from “Cilione Antonino s.r.l.”, Reggio Calabria, Italy. The furocoumarin-free formulation avoids phototoxicity, according to the assessment report of the EMA, HMPC (13 September 2011 EMA/HMPC/56155/2011). According to the information provided by the manufacturer, citrus fruits were peeled, and the peels were chopped and minced thoroughly. The obtained pre-processed peels were collected and subjected to vacuum distillation by immersion in a heated oil bath. The essential oil was then separated from the aqueous phase and dried in order to reduce the final volume to the desired amount.

### 4.2. Gas Chromatography-Mass Spectrometry (GC-MS)

Gas chromatography-mass spectrometry (GC-MS) was carried out on tested samples through a Hewlett-Packard 6890 gas chromatograph with an SE-30 capillary column 100% dimethylpolysiloxane (30 m length, 0.25 mm in diameter and 0.25 µm film thickness) equipped with a Hewlett Packard 5973 mass spectrometer. Analyses were performed by using a programmed temperature from 60 to 280 °C, with a rate of 16 °C/min and helium as carrier gas (linear velocity 0.00167 cm/s). Compounds were identified by matching spectra with those of the Wiley 138 mass spectral library [51].

### 4.3. Varroa Destructor: Toxicity Evaluation

The experiments were carried out in the parasitology laboratory of Centro Interdipartimentale Servizi Veterinari Salute Umana e Animale (CIS VetSUA), Department of Health Sciences, Magna Graecia University of Catanzaro.

A residual bioassay, which has been widely used to detect toxicity and pesticide resistance in arthropods [52,53,54,55], was used to evaluate the acute toxicity of 5 potential natural varroacides. The essential oils tested were bergamot (*C. bergamia*), grapefruit (*C. paradisi*), lemon (*C. limon*), orange (*C. sinensis*), and mandarin (*C. reticulata*). The method used was adapted from Gashout and Guzmán-Novoa [41], with minor adjustments.

The 5 essential oils used, plus Amitraz, were diluted in HPLC grade acetone up to a concentration of 2 mg/mL, 1 mg/mL, and 0.5 mg/mL. Amitraz (Merck, 45323) and acetone were used as positive and negative controls, respectively.

The experiments were carried out in April and May 2021. Various apiaries in the province of Catanzaro, infested with *V. destructor* and which had not been treated with acaricides in the previous six months, were used as sources of mites.

Each experimental replicate started with the mites’ collection (around 100–200 mites per time) from the same apiary obtained from a drone trap frame and specifically from non-pigmented pupae. The mites were removed from the colonies of origin and transported to the laboratory.

Each Eppendorf tube (1.5 mL) was previously filled with 50 µL of acetone-diluted essential oil at the different concentrations and placed opened in the oven for acetone evaporation. The tubes were frequently rolled up on their walls to coat the walls and to facilitate the acetone to evaporate and leave the EOs in the walls of the tube. For each technical replicate (for each oil and for each positive and negative control), five live adult female mites were gently transferred inside the previously prepared tube with a small paintbrush. Once the mites were transferred inside, the tubes were closed tightly and placed in a dark room at 34 °C and 65% relative humidity. For each essential oil and the negative and positive control, this procedure was repeated 14 times (14 technical replicates) for the lowest concentration of 0.5 mg/mL and 7 times (7 replicates) each for the concentrations of 1 mg/mL and 2 mg/mL.

The relative acute toxicity of each product was determined by recording mite mortality after 1 h exposure to each treatment. This time was chosen to determine the acute toxicity independent of mite mortality due to other causes. This was because mites are sensitive to artificial environments when kept for more than 4 h out of their natural habitat and may suffer from starvation and water loss [56]. One hour after exposure, the parasites were transferred from the tubes to a Petri dish and examined under a stereoscopic microscope. Mites that did not move when probed with a fine paintbrush were considered dead. Mites that moved one or more legs were classified as inactive. After one hour of exposure, the inactive mites were transferred from the tube with essential oils to a new clean tube in controlled environmental conditions and observed for another hour to verify death or a possible recovery. In all cases, there was no recovery of the exposed mites that stayed in a steady condition. The *V. destructor* identification was performed by microscopic manual observation.

The statistical analysis was performed with jmpSAS software for the calculation of the groups average. Student’s *t* test was applied to evaluate the statistical significance of the difference between the control group and the treated groups.

### 4.4. Honey Bee Workers: Toxicity Evaluation

According to the obtained results, the most promising essential oils (bergamot and lemon) were tested for toxicity on the adult honey bee. Three technical replicates were performed. 

For each replicate, a total of twenty *A. mellifera* worker bees were randomly collected. The honey bees came from a larger group, where honey bees were collected from different frames and randomly mixed [57]. Three 50 mL Falcon tubes were filled with 1.6 mL of acetone, bergamot, and lemon essential oil at the same concentrations as for the *V. destructor* viability tests and were rolled over the walls several times to coat the walls with liquid and allow the acetone contained in the solution to evaporate. Once dried, five individuals for each essential oil and acetone as negative control were transferred inside the tube. 

The Falcon tubes with the honey bees inside were stored in a dark room at 34 °C and 65% R.H. for one hour.

After 1 h, the honey bees were transferred from the tubes to the cages. The cages consisted of a cylindrical plastic container. Two 1.5 mL Eppendorf tubes were placed on the side walls of the cages as feeding devices. Three holes had previously been drilled into the Eppendorf tubes, and a 30% solution of glucose and water was inserted into each of them, respectively. Mortality was verified at 24 and 48 h.

## 5. Conclusions

Our study suggests the good in vitro efficacy of *Citrus* fruits essential oils for the neutralization of the parasitic mite *V. destructor.* This result, together with the chemical characterization and with the positive result about the toxicity on honey bees, drives the way towards the further steps for the in vivo evaluation of these EOs. Although the results are encouraging, it must be emphasized that other in vitro tests performed by other research groups were often not consistent with the results obtained in the field tests. Numerous factors can affect the overall effectiveness of essential oils under field conditions. Among those are the method of delivery, the duration of treatment, the environment of the colony (presence or absence of brood), and the ambient temperature. Formulations that delay the evaporation rates and provide a controlled release, independent of external factors, are therefore necessary. Microencapsulation or gel formulations may be able to ensure a more constant release of these volatile compounds. 

With reference to the costs, considering that bergamot EO price is around EUR 80 per kilogram, and considering that in vitro it is already effective at the concentration of 0.5 mg/mL, it can be easily calculated that its application would be extremely cost-effective.

In conclusion, this study represents another branch of green veterinary pharmacology (GVP) that pushes the research towards the discovery and use of natural preparations rather than single synthetic molecules whose persistent use and accumulation in the environment might make them more susceptible to resistance mechanisms.

## Figures and Tables

**Figure 1 pathogens-10-01182-f001:**
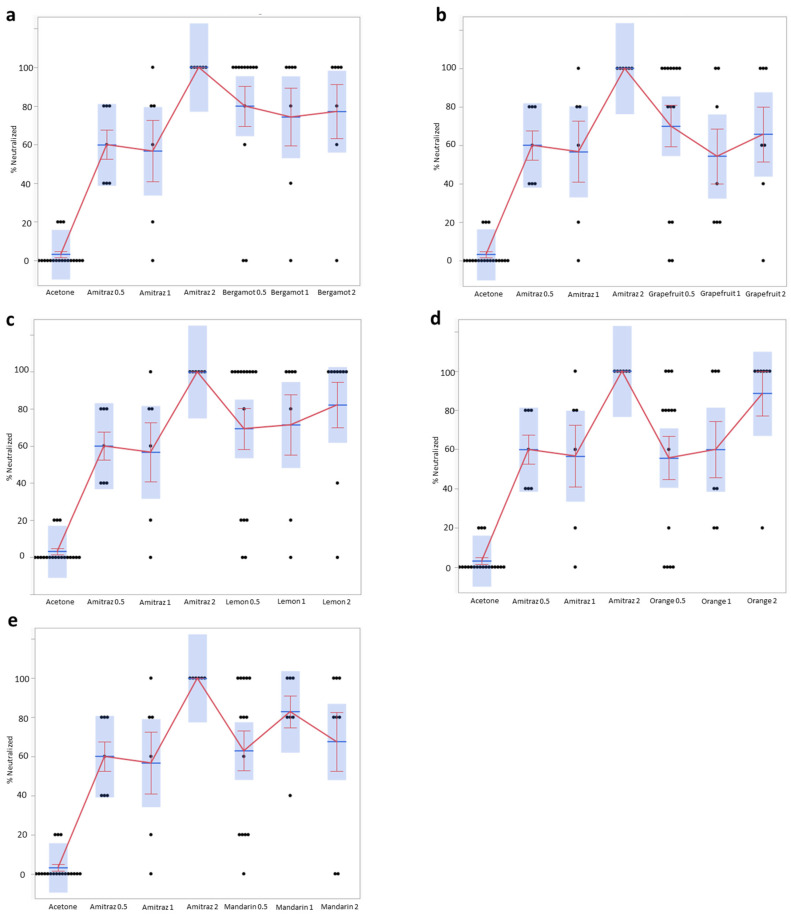
Representation of the percentages of neutralized mites for bergamot (**a**), grapefruit (**b**), lemon (**c**), orange (**d**), and mandarin (**e**) essential oils used at 0.5 mg/mL, 1 mg/mL, and 2 mg/mL in comparison with Amitraz. *p* value is ≤ 0.001 for any EO at each concentration in comparison with acetone (standard error indicated with the red bars).

**Table 1 pathogens-10-01182-t001:** Chemical characterization of the Eos.

N.	Compound ^(a)^	Rt ^(b)^			RAP ^(c)^		
			Mandarin	Lemon	Bergamot	Orange	Grapefruit
1	Thujene	6.197	3.40 ± 0.10	2.22 ± 0.27	1.05 ± 0.10	3.40 ± 0.42	-
2	Alpha-pinene	6.363	5.32 ± 0.42	7.97 ± 1.00	3.21 ± 0.34	5.32 ± 0.60	4.00 ± 0.36
3	Camphene	6.666	−	0.49 ± 0.04	0.14 ± 0.01	−	−
4	Beta−pinene	7.254	−	19.31 ± 1.56	9.61 ± 1.17	−	−
5	Sabinene	7.266	6.77 ± 0.23	−	−	6.77 ± 0.30	3.28 ± 0.39
6	Beta-myrcene	7.517	8.00 ± 0.34	4.50 ± 0.38	2.18 ± 0.14	8.00 ± 1.00	6.29 ± 0.78
7	Phellandrene	7.746	−	0.25 ± 0.01	−	−	−
8	Limonene	8.169	47.85 ± 3.41	35.21 ± 3.67	25.17 ± 2.77	47.85 ± 5.68	47.62 ± 5.87
9	Gamma-terpinene	8.712	1.87 ± 0.15	8.39 ± 0.98	−	1.87 ± 0.07	−
10	Linalool oxide	8.838	−	−	1.79 ± 0.21	−	−
11	Alpha-terpinolene	9.049	−	0.58 ± 0.04	−	−	−
12	Linalool	9.209	2.58 ± 0.19	0.26 ± 0.02	11.32 ± 1.33	2.58 ± 0.20	1.26 ± 0.11
13	P-mentha-2,8-dien-1-ol	9.495	1.32 ± 0.15	−	0.30 ± 0.01	1.32 ± 0.12	1.26 ± 0.12
14	Lemongrass	9.815	−	−	−	−	0.29 ± 0.03
15	Cis-Carveol	10.615	1.61 ± 0.20	0.32 ± 0.01	−	1.61 ± 0.17	1.63 ± 0.10
16	Trans-Carveol	10.792	−	−	0.86 ± 0.10	1.33 ± 0.09	1.35 ± 0.17
17	Linalyl acetate	10.918	4.44 ± 0.45	−	13.50 ± 1.95	4.44 ± 0.37	−
18	Citral	11.089	0.61 ± 0.03	4.14 ± 0.37	−	0.61 ± 0.06	0.99 ± 0.10
19	Geranyl acetate	12.084	0.47 ± 0.02	−	−	0.47 ± 0.03	1.12 ± 0.14
20	Trans-Caryophillene	12.558	0.18 ± 0.01	Tr ^(d)^	−	0.18 ± 0.01	1.77 ± 0.22
21	Uroterpenol	12.680	−	Tr	−	−	−
22	Valencene	13.147	0.29 ± 0.01	−	−	0.29 ± 0.03	−
23	Nerolidol	13.593	0.14 ± 0.01	−	−	0.14 ± 0.01	−
24	Beta-bisabolene	16.502	0.19 ± 0.01	−	−	0.19 ± 0.01	0.19 ± 0.01
25	Neoisolongifolene-8,9-dehydro	18.108	0.22 ± 0.02	−	−	0.22 ± 0.01	−

^(a)^: Compounds listed in order of elution from SE30 MS column. ^(b)^: Retention time (min). ^(c)^: Relative area percentage (peak area relative to total peak area in total ion current %). ^(d)^: Tr (value lower than 0.1% are reported as traces). Data are reported as mean ± S.D. (n = 3) of three independent experiments.

**Table 2 pathogens-10-01182-t002:** *V. destructor* neutralized percentage and SD (±) after the treatment with *Citrus* spp. Essential oils, acetone, and Amitraz.

Concentration (mg/mL)	Bergamot (*C. bergamia*)	Grapefruit (*C. paradisi*)	Lemon (*C. limon*)	Orange (*C. sinensis*)	Mandarin (*C. reticulata*)	Acetone (*− control*)	Amitraz (*+ control*)
0.5 mg	80 ± 37	70 ± 40	69 ± 43	56 ± 42	63 ± 38	3 ± 7	60 ± 20
1 mg	74 ± 39	54 ± 38	71 ± 43	60 ± 38	83 ± 21		57 ± 39
2 mg	77 ± 37	65 ± 38	82 ± 37	89 ± 30	67 ± 43		100 ± 0

## Data Availability

No extra supporting data are needed for this manuscript.

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
