# Peer review of "In Vitro Evaluation of Acute Toxicity of Five *Citrus* spp. Essential Oils towards the Parasitic Mite *Varroa destructor"

_pathogens, 2021, doi:10.3390/pathogens10091182_

Round 1

Reviewer 1 Report

General comments

In general, this manuscript has a valuable topic. The manuscript is well written except for moderate English language check required. The experimental design is adequate. There are some minor comments.

In general, please avoid using personal pronouns such as we, our results, and apply this rule throughout the manuscript (for example Line 215: In this work we and in line 246 : Our study).

Abstract:

The aim of the study was clearly stated, and the abstract was well written except for some minor English mistakes as it is found in the entire manuscript. But in terms of writing style and English sound it is ok.

Key words; Please add In Vitro toxicity to the keywords list.

Introduction:

This section is well written and provides enough background.

Results:

-The results were well presented but with poor discussion.

Conclusion:

This section needs some more work for example, line 194-203: Please add the proper citation

I had a hard time to understand this section I think this was due to two reasons. First one is that the authors did provide enough discussion to the results. Second reason is the language. Please rewrite this section.

Materials and Methods:

The experimental design was suitable and adequate to the current study.

Conclusion:

This section provides a good conclusion for the study and includes the significant findings with some recommendations for further investigations on this topic. The conclusion was supported by the results

-. But the way it is written is difficult to understand. Please check the English.

-Also move the conclusion in its right place in the manuscript right before the references.

References:

The authors provided enough citations, and it was UpToDate.

*I find that the manuscript is scientific point of view with some extra effort the manuscript will be improved and suitable for publication.

Author Response

We would like to thank the reviewer for the precious effort that helped to improve the manuscript.

General comments

In general, this manuscript has a valuable topic. The manuscript is well written except for moderate English language check required. The experimental design is adequate. There are some minor comments.

In general, please avoid using personal pronouns such as we, our results, and apply this rule throughout the manuscript (for example Line 215: In this work we and in line 246 : Our study).

Response: we would like to thank the reviewer for this comment, all the personal forms have been removed from the manuscript.

Abstract:

The aim of the study was clearly stated, and the abstract was well written except for some minor English mistakes as it is found in the entire manuscript. But in terms of writing style and English sound it is ok.

Response: many thanks for this comment, the abstract has been modified as suggested.

Key words; Please add In Vitro toxicity to the keywords list.

Response: We would like to thank the reviewer for the advice, we added the suggested keyword which now helps with providing a better snapshot of our research.

Introduction:

This section is well written and provides enough background.

Results:

-The results were well presented but with poor discussion.

Response: in accordance with the reviewer's comment, we have improved and revised the results section accordingly.

Conclusion:

This section needs some more work for example, line 194-203: Please add the proper citation

Response: We would like to thank the reviewer for this comment, we have added the bibliographic references related to the highlighted text and deleted a sentence.

I had a hard time to understand this section I think this was due to two reasons. First one is that the authors did provide enough discussion to the results. Second reason is the language. Please rewrite this section.

Response: the entire section was revised accordingly.

Materials and Methods:

The experimental design was suitable and adequate to the current study.

Conclusion:

This section provides a good conclusion for the study and includes the significant findings with some recommendations for further investigations on this topic. The conclusion was supported by the results

-. But the way it is written is difficult to understand. Please check the English.

Response: in accordance with the comment, we have improved the English and revised the section.

-Also move the conclusion in its right place in the manuscript right before the references.

Response: many thanks for this useful comment, the conclusion section has now been moved in the correct place.

References:

The authors provided enough citations, and it was UpToDate.

*I find that the manuscript is scientific point of view with some extra effort the manuscript will be improved and suitable for publication.

Overall response: we would like to thank the referee for the effort dedicated to this revision and we believe that the comments were very spot on and really helped with improving the manuscript.

Reviewer 2 Report

I give you my corrects in the pdf attavhed.

I liked your study but we have to continue working in it. Good preliminary results.

Author Response

Line 30 In my opinion, It would be better a minimum of 10 replicates to obtain more reliable results.

Response: we apologise with the referee for this misunderstanding outcoming from our poor writing in the first version. 5 mites were used as one technical replicate for each oil/drug for each experiment. Actually, every experiment was carried out with 35 mites and, for the tests of the concentrations at 0.5, 14 technical replicates were performed (490 mites). Many thanks to the referee for spotting this imprecision that now helped with amending the manuscript.

Line 133 If you continue with this sctructure, pay attention to this phase. It seems like you already talked about methods.

Response: we would like to thank the reviewer for this comment, we have changed the text as advised.

Line 133 In my opinion, 5 mites are very few to obtain reliable results. Minimum 10 mites.

Response: we apologise with the referee for this misunderstanding outcoming from our poor writing in the first version. 5 mites were used as one technical replicate for each oil/drug for each experiment. Actually, every experiment was carried out with 35 mites and, for the tests of the concentrations at 0.5, 14 technical replicates were performed (490 mites). Many thanks to the referee for spotting this imprecision that now helped with amending the manuscript.

Line 140 How many mites did you used in total? Materials and Methods is very confused and finally I do not know the sample size.

Response: all the precise numbers have now been described in the results section (2.1). Moreover, the whole Material and Methods section has been revised.

Line 143 I think that It is better If you put in Y axis the % like in Table 1 instead of 1, 2, 3, 4 and 5.

Response: Thanks for this comment, it is definitely more logical to indicate the percentage of inactivated parasites in the y axis.

Line 145 in the graphic you put neutralized.

Response: now amended.

Line 147 1c instead of 2c?

Response: many thanks for spotting this mistake, now amended.

Line 159 I miss the results of orange EO. You said in abstract that you tested it.

Response: thanks for this comment, Orange results are showed in figure 1, but not in figure 2. Results related to orange essential oil were not represented in figure 2 because it is the least effective.

Line 163 Graphic of orange EO?

Response: thanks for this comment, Orange results are showed in figure 1, but not in figure 2. Results related to orange essential oil were not represented in figure 2 because it is the least effective and we didn’t think it was wise to overcrowd the figure.

Line 167 In my opinion, 5 are very few.

Response: we apologise with the referee for this misunderstanding outcoming from our poor writing in the first version. 5 mites were used as one technical replicate for each oil/drug for each experiment. Actually, every experiment was carried out with 35 mites and, for the tests of the concentrations at 0.5, 14 technical replicates were performed (490 mites). Many thanks to the referee for spotting this imprecision that now helped with amending the manuscript.

Line 170 Be careful! Mortality is not the unique important thing It could be affect to other parameters like rate of ovoposition...

Response: in this case, only worker bees were tested for toxicity. Oviposition rate could not be assessed. Although we totally agree, mortality is not satisfactory as parameter. We specified in the text that these results will be followed by the  in vivo experiments in order to evaluate all other parameters. Many thanks for this suggestion.

Line 173 Are you sured? In bergamot are 11 compounds.

Response: many thanks for this comment, correct 11 compounds detected at least. Now amended in the text.

Line 176 And what happens with this fact? You propose this compound as repelent?

Response: many thanks to the referee for this comment. All authors strongly agree with the referee about the importance of this compound. The discussion section (lines 253-255) has now been improved thanks to this comment.

Lines 177-181 In Figure 2, all EOs seem to be effective (more or less depending on the concentration) althought less than Amitraz. So I undestand that all EOs tested contain some effective compound. As you said in lines 174 and 175, they share 4 compounds. You talked about limonene and tried to explain the values of bergamont. What do you think about  Linalool? It is common in all four and, it has a high value in bergamont and in the rest could be sinergyc with other compounds.

Response: many thanks to the referee for this comment. All authors strongly agree with the referee about the importance of Linalool. The discussion section (lines 253-255) has now been improved thanks to this comment.

Line 241 In my opinion, suggests. Does not confirm. The results are very preliminary and the sample size used generates many doubts.

Response: even though there was probably a misunderstanding about the number of samples/replicates analysed, we agree with the reviewer and we chose to switch to the more careful form as suggested. Thanks for the comment.

Line 253 Acronym unknown

Response: many thanks for this useful comment. Now specified.

Line 256 (results section) It is very confused. It needs to be clarified and restructured.

Response: many thanks for this comment, the whole section has now been rewritten.

Line 257 In the abstract you said that you did 6 replicates. In this part you do not explain that.

Response: many thanks for this comment, the whole section has now been partially rewritten. The technical replicates were much more (14 for the 0.5 mg/mL concentration). All the sections with such a mistake were rewritten.

Line 298 Questionable

Response: many thanks for this comment, agreed, the sentence has been amended.

Line 305 If you change the structure: According to the chemical results obtained...

Response: Structure changed.

Line 305 mandarin, grapefruit and orange?

Response: only the most promising EOs were tested for the toxicity evaluation because, most probably, those will be the only ones moving to the in vivo tests.

Line 325 Did you identify the mites to confirm that they are V. destructor and not an other species?

Response: many thanks to the referee for this comment. Varroa destructor is the mite specie that is diffused in this area. However, even if we morphologically identified the Varroa genre, it is more precise to talk about Varroa spp. rather than talking about the V. destructor specie. We amended the whole manuscript accordingly

Reviewer 3 Report

This study tried to evaluate the potential effects of Essential oils from Citrus spp against Varroa destructor.

The manuscript, despite the topic could be interesting, does not fully demonstrate what is claiming. In addition, the experimental design, in general, materials and methods are not properly described. The results should be reconsidered and the discussion needs substantial improvements.

English must be improved.  It was not possible to understand several sentences.

1) Title: all tested EOs derive from Citrus spp. which are not only cultivated in Calabria - therefore it is not appropriate to indicate this region in the title. it is also claimed in several parts of the text that the term Calabria ina way that means that it seems that onluy Calabrian Citrus spp would be effective.

2) Abstracat should be sustantially improved.

3) It is not know why the authors start M&M with a chemical characterization of the EOs and then the results of these activities are reported as last part of the results. It is also not very clear the meaning of this chemical analysis section. The chemical investiagtion seems not very relevant in this context.

4) M&M does not include any section related to data analysis / statistical analysis. Actually no statistical analysis of the data have been reported. How is it possible to claim differences between the types of EOs /cotrol etc. ?

5) The number of mites analysed in each trial (5) is too low to obtain statistically sound results.

6) In addition, the way in which the mites have been challenged is not common. The experimental design with the controls is not fully described to be able to understand it.

7) It is not clear how the EOs have been prepared 

Author Response

This study tried to evaluate the potential effects of Essential oils from Citrus spp against Varroa destructor.

The manuscript, despite the topic could be interesting, does not fully demonstrate what is claiming. In addition, the experimental design, in general, materials and methods are not properly described. The results should be reconsidered and the discussion needs substantial improvements.

Response: the whole manuscript has been substantially revised as requested paying major attention to the results and discussion sections.

English must be improved.  It was not possible to understand several sentences.

Response: the writing of the entire manuscript has been revised.

1) Title: all tested EOs derive from Citrus spp. which are not only cultivated in Calabria - therefore it is not appropriate to indicate this region in the title. it is also claimed in several parts of the text that the term Calabria ina way that means that it seems that onluy Calabrian Citrus spp would be effective.

Response: Many thanks to the referee for spotting this imprecision. All the sentences where Calabrian region was mentioned have been amended trying to explain that the manufacturing was done in Calabria and with Calabrian fruits only. Of course, this does not mean that other citrus fruits won’t be effective. Although, bergamot, for example is cultivated mainly in Calabria.

2) Abstracat should be sustantially improved.

Response: many thanks to the referee for this comment. The abstract has been totally rewritten.

3) It is not know why the authors start M&M with a chemical characterization of the EOs and then the results of these activities are reported as last part of the results. It is also not very clear the meaning of this chemical analysis section. The chemical investiagtion seems not very relevant in this context.

Response: many thanks to the referee for this useful comment. Although, while we completely agree with the first part of the comment, we all strongly disagree with the second. The order of the M&M section has been modified as suggested. However, we all believe that the qualitative chemical characterization is important for this paper because it provides the composition of the EOs here prepared (i) and provides the composition typical Calabrian EOs as bergamot.

4) M&M does not include any section related to data analysis / statistical analysis. Actually no statistical analysis of the data have been reported. How is it possible to claim differences between the types of EOs /cotrol etc. ?

Response: many thanks for this comment. The number of mites and technical replicates has now been indicated and the section about statistical analysis has now been described in M&M.

5) The number of mites analysed in each trial (5) is too low to obtain statistically sound results.

Response: we apologise with the referee for this misunderstanding outcoming from our poor writing in the first version. 5 mites were used as one technical replicate for each oil/drug for each experiment. Actually, every experiment was carried out with 35 mites and, for the tests of the concentrations at 0.5, 14 technical replicates were performed (490 mites). Many thanks to the referee for spotting this imprecision that now helped with amending the manuscript.

6) In addition, the way in which the mites have been challenged is not common. The experimental design with the controls is not fully described to be able to understand it.

Response: Many thanks to the referee for this comment, we followed the method described by Gashout and Guzmán-Novoa (DOI: 10.3896/IBRA.1.48.4.06). This source has now been cited in the text.

7) It is not clear how the EOs have been prepared

Response: many thanks for this comment. EOs were purchased from the company that has a standardized preparation protocol. We briefly indicated the method which should be sufficient. We hope this will satisfy the referee’s request.

Round 2

Reviewer 2 Report

Improved version. Some details remain.

Author Response

Line 164 In my opinion, it is not a good reason to not be represented. The difference with Amitraz are only 4.

Response

Representation of orange EO has been added to figure 1. Figure 2 was merged with figure1 as suggested by a referee. The text has been amended accordingly.

All other minor comments have been directly amended in the text without providing a reply. Many thanks for the effort to improve the manuscript.

Reviewer 3 Report

The manuscript has been improved compared to the previous version. However, there are still several issues that should be clarified and improved.

It is not clear why the authors changed Varroa destructor with Varroa spp. Actually, I would expect that they tested Varroa destructor (or V. destructor, after the first use) and not other species of the Varroa genus.

In addition, please use always the term honey bee to substitute the term bee if referrred to Apis mellifera

This should be changed back.

English is still not very accurate and several typos are still in the text:

 a few examples: 

line 25 - genus and not genre

line 26 - common and not commont

line 37 - Calabrian and not calabrian (even if it would not be appropriate to use Calabrian bergamot)

Please go through the whole text and check carefully all typos.

English should be further improved in several other sentences

In the text, where the words Varroa or varroa are used - it would be better to substitute them with mite or mites according to the singular or plural use - example: line 29 and in several other parts

Acetone or acetone?

Abstract

It is not clear how the EOs have been tested and the experimental design/test

Introduction

line 48 honey bee farming

lines 51-52 species ... honey bee colonies ...levels

Results

This paragraph is confusing

First the authors should present the chemical characterization of the EOs. Actually, one subparagraph for the characterization of the EOs is missed. Different preparations of EOs are not the same - that means that first this part should be presented in order to clarify what the authors are using

The chemical characterization is only based on GC-MS - this is a limit of the study.

No s.d. has been reported in the different components (RAP) that means that analyses have been just run once per sample? this should be clarified 

Lines 132-133 - it is not really clear this sentence - the reference has been also cited in a wrong order

Lines 277-278 - it should be detailed a little bit more in oder to clarify the experimental design also here.

Figure 1a - Acetone is indicated twice

Please simplify the legend of Figure 1 - the text is repeated several times even if not needed - only different concentrations were used to distinguish the figures - combination of the information in just one figure whould be good to have a quick overview of differences between concentrations and EOs

Figure 2 reports similar information than in Figure 1- again just one figure could be combined 

The graphical quality of the two figures is quite poor - the legend should be improved to clarify the content and what the spot means and what the different lines means - information on the statistical differences should be included in these figures or in Table 1.

Table 2. (that should be moved when the paragraph is moved and should become table 1). It missed the legend - sd should be reported

Discussion: the chemical characterization - even if limited should be moved at the beginning.

Information on the common concentration used for Amitraz in field applications should be reported for the discussion

The discussion should be also report information on the cost - or potential cost of use of EOs in field applications 

The limits of this study should be evaluated: the in vitro test is different from in vivo test 

M&M

lines 266-267: It is not yet clear how EOs have been produced - please better define the protocol/steps used - for example, what vegetable matrices were use, how long the different steps last - different protocols have been proposed for the preparation of EOs

The chemical characterization of the EOs should be after the section 4.1 - in this way the material used for the tests would be well characterized

Line 279: Amitraz - please indicate the supplier 

Lines 284-286: please indicate the periods of the year of collection of the mites - this could make some differences

line 289: 1.5 (please check all other numbers with decimals - dot and not comma)

line 291: not clear what is in brackets

line 295: R.H. please expand

Many other details should be improved - the way in which numbers are written, and so on. Plese be very precise in the writing and in the way data are presented.

Author Response

The manuscript has been improved compared to the previous version. However, there are still several issues that should be clarified and improved.

It is not clear why the authors changed Varroa destructor with Varroa spp. Actually, I would expect that they tested Varroa destructor (or V. destructor, after the first use) and not other species of the Varroa genus.

Response

Varroa destructor mites were employed in this study. The morphological identification was manually performed via observation. We erroneously switched to Varroa spp. after the first referral step following a comment from a referee about the identification of the mites. We did not genetically identify the mites because there is no possibility that other species than Varroa destructor are present in this geographical area. For further reading please take a look at the following scientific evidence “Divergent evolutionary trajectories following speciation in two ectoparasitic honeybee mites” (https://www.nature.com/articles/s42003-019-0606-0). Many thanks for this comment and, please accept our apologies for the double change.

In addition, please use always the term honey bee to substitute the term bee if referrred to Apis mellifera

This should be changed back.

Response

All the points of the manuscript were we referred to Apis mellifera have been modified from “bees” to “honey bees”.

English is still not very accurate and several typos are still in the text:

a few examples:

line 25 - genus and not genre

Response

Many thanks for this comment. The term has been amended as suggested.

line 26 - common and not commont

line 37 - Calabrian and not calabrian (even if it would not be appropriate to use Calabrian bergamot)

Please go through the whole text and check carefully all typos.

Response

We would like to thank the reviewer for this comment. All the typos here mentioned have been corrected.

English should be further improved in several other sentences

In the text, where the words Varroa or varroa are used - it would be better to substitute them with mite or mites according to the singular or plural use - example: line 29 and in several other parts

Response

Many thanks for this advice, the terms have been replaced as advised.

Acetone or acetone?

Response: the typo highlighted has been corrected.

Abstract

It is not clear how the EOs have been tested and the experimental design/test

Response

We improved the abstract trying to be as clear as possible. We hope it is now clear enough.

Introduction

line 48 honey bee farming; lines 51-52 species ... honey bee colonies ...levels

Response

We would like to thank the reviewer for this comment. The typos highlighted have been corrected.

Results

This paragraph is confusing

First the authors should present the chemical characterization of the EOs. Actually, one subparagraph for the characterization of the EOs is missed. Different preparations of EOs are not the same - that means that first this part should be presented in order to clarify what the authors are using

Response

Many thanks for this comment. The EOs characterization paragraph has been moved to the beginning of the results section as suggested. This helped to be more clear in the whole results section.

As per the second part of this comment, we agree with the referee that we missed the description of the methods used for the EOs preparation. A section about this topic has been added as requested. Please look below at the answer about the comment regarding M&M and at the improved M&M section.

The chemical characterization is only based on GC-MS - this is a limit of the study.

Response

According to our knowledge and to other studies (see below), GC-MS should represent the correct approach to study the composition of EOs that carry mainly volatile compounds. In our opinion, this should not represent a limitation of this study.

https://pubs.acs.org/doi/10.1021/jf051922u

https://www.sciencedirect.com/science/article/pii/S222116911630404X

https://www.hindawi.com/journals/bri/2020/8861798/

No s.d. has been reported in the different components (RAP) that means that analyses have been just run once per sample? this should be clarified

Response

Thanks for this comment. The quantitative analysis of the essential oil components was performed in triplicate. We added the standard deviation values ​​to the table as suggested.

Lines 132-133 - it is not really clear this sentence - the reference has been also cited in a wrong order

Response: many thanks to the referee for this comment. The sentence was amended and the bibliographic note was cited correctly.

Lines 277-278 - it should be detailed a little bit more in oder to clarify the experimental design also here.

Response

We would like to thank the reviewer for this comment. The part subsequently described provides a good resume of the method used.

Figure 1a - Acetone is indicated twice

Response

Thanks for spotting this mistake. Now amended.

Please simplify the legend of Figure 1 - the text is repeated several times even if not needed - only different concentrations were used to distinguish the figures - combination of the information in just one figure whould be good to have a quick overview of differences between concentrations and EOs

Figure 2 reports similar information than in Figure 1- again just one figure could be combined

The graphical quality of the two figures is quite poor - the legend should be improved to clarify the content and what the spot means and what the different lines means - information on the statistical differences should be included in these figures or in Table 1.

Response

Many thanks for this comment. The mentioned figure has been merged and improved accordingly.

Table 2. (that should be moved when the paragraph is moved and should become table 1). It missed the legend - sd should be reported

Response

Table 2 has now been moved at the beginning of the results section. We would like to thank the referee for this comment that helped with a more logical presentation of the results.

Discussion: the chemical characterization - even if limited should be moved at the beginning.

Response

We would like to thank the reviewer for this comment. As suggested, we moved the chemical characterization to the beginning.

Information on the common concentration used for Amitraz in field applications should be reported for the discussion

Response

The discussion section has been improved as suggested indicating the Amitraz concentration used in the strips for honey bees in vivo treatment.

The discussion should be also report information on the cost - or potential cost of use of EOs in field applications

Response

Many thanks for this comment. An estimation of the costs for field application was added in the conclusions section.

The limits of this study should be evaluated: the in vitro test is different from in vivo test

Response

We would like to thank the reviewer for this comment. In the conclusions section, the limitations of in vitro tests and the differences in the results that could be obtained in the field in relation to various factors have been explained.

M&M

lines 266-267: It is not yet clear how EOs have been produced - please better define the protocol/steps used - for example, what vegetable matrices were use, how long the different steps last - different protocols have been proposed for the preparation of EOs

Response

Many thanks for this comment. We modified this section and provided the requested details accordingly.

The chemical characterization of the EOs should be after the section 4.1 - in this way the material used for the tests would be well characterized

Response

The section has been moved accordingly.

Line 279: Amitraz - please indicate the supplier

Response

Thanks for this comment. Supplier indicated.

Lines 284-286: please indicate the periods of the year of collection of the mites - this could make some differences

Response

Many thanks to the referee for this comment. As suggested, the period of the year when mites were collected has now been indicated.

line 289: 1.5 (please check all other numbers with decimals - dot and not comma)

Response: many thanks for the comment. We have corrected this typo.

line 291: not clear what is in brackets

Response

Thanks for this comment. Brackets and their content has been removed since it was not necessary.

line 295: R.H. please expand

Response

The label was removed and indicated in the full form.

Many other details should be improved - the way in which numbers are written, and so on. Plese be very precise in the writing and in the way data are presented.

All sections have been improved accordingly. We now hope that the referee is satisfied with the improvements.

Round 3

Reviewer 3 Report

The manuscript has been substantially improved. English is still poor in a few parts.